# 3D fluorescence staining and confocal imaging of low amount of intestinal organoids (enteroids): Protocol accessible to all

Ami Gloria Toulehohoun[1]*, Caroline Bouzin[2], Aurélie Daumerie[2], Luca Maccioni[3], Peter Stärkel[1,4]

**1** Laboratory of Hepato-Gastroenterology, Institut de Recherche Expérimentale et Clinique, Université Catholique de Louvain, Brussels, Belgium, **2** IREC Imaging Platform Belgium (2IP, RRID:SCR_023378), Institut de Recherche Expérimentale et Clinique, Université Catholique de Louvain, Brussels, Belgium, **3** Laboratory of Liver Diseases, National institute of Alcohol Abuse and Alcoholism (NIAAA), The National Institutes of Health (NIH), Rockvile, MD, United States of America, **4** Department of Hepato-Gastroenterology, Cliniques Universitaires Saint-Luc, Brussels, Belgium

◎ These authors contributed equally to this work.

* ami.toulehohoun@uclouvain.be

## Abstract

The emerging field of 3D organ modeling encounters several imaging issues in particular related to antigen retrieval and sample loss during staining processes. Due to their compact shape, several antibodies fail to penetrate intact organoids or spheroids. Histology of organoids can be approached by paraffin inclusion and sectioning at 5 μm as performed for biopsies. However, to fully understand organoid behavior, including cellular organization, extracellular matrix structure, and their response to treatments, 3D imaging is essential. Here we propose an easy workflow allowing (1) immunostaining with a HIER step, (2) preservation of the intact shape of the organoids, (3) sample immobilization in a focal plane reachable for high resolution/short working distance lenses, and (4) minimizing the risk of loss of precious material.

## Introduction

Organoids are 3D *in vitro* models, mimicking the native patient organ (healthy or pathologic) in terms of structural and functional aspects. They offer unparalleled potential for disease modelling and large-scale therapeutic screening [1]. Compared to animal models, organoids derived from patients' cells translate into greater similarity in physiological and disease processes and offer a more ethically responsible approach.

Following proper characterization and *in vitro* expansion, these "mini-tissues" can be established as biobanks, serving as valuable research resources for biomarker identification and novel therapeutic target development. Furthermore, organoid models hold particular promise for studying rare pathologies, where limited access to biological material delays research progress. Their versatility extends to diverse areas, including tumor biology, inflammatory diseases, regenerative medicine, and more. Importantly, unlike formalin-fixed paraffin-embedded

**Data Availability Statement:** All data files are available from the Protocol.io database (DOI: 10. 17504/protocols.io.kqdg323m7v25/v1).

**Funding:** The author(s) received no specific funding for this work.

**Competing interests:** The authors have declared that no competing interests exist.

tissues, these "living biobanks" provide a source of fresh material suitable for a wide range of analyses, including histological, protein expression, genomic, and metabolomic studies.

Histology of organoids can be approached by paraffin inclusion and sectioning at 5 μm as performed for biopsies [2]. However, to fully understand organoid behavior, including cellular organization, extracellular matrix structure, and their response to treatments, 3D imaging is essential. Different imaging technologies allow for 3D imaging of these small (50–300 μm diameter) structures, like light sheet microscopy and confocal microscopy. Light sheet microscopy offers high imaging speed and good spatial resolution [3–5] while confocal microscopy is the most widely accessible device. Although sample positioning is challenging on both devices, organoid inclusion in agarose sticks for light sheet microscopy leads to a massive sample loss. For this reason, we focused on confocal microscopy.

The emerging field of 3D organ modeling requires adequate sample preparation in order to generate faithful representations of the real 3D structure. Heat induced epitope retrieval (HIER) is routinely used for 2D Formalin-Fixed Paraffin-Embedded (FFPE) sections. However, this technique cannot be extrapolated to 3D staining. To the best of our knowledge, HIER is not well established for 3D staining due to the complexity of the samples management.

First, fixation with formaldehyde is known to create crosslinking by forming covalent bonds between proteins [6]. This network can also be an obstacle to antibodies penetration. On 2D sections, HIER (acidic or basic pH) is commonly performed to break this network and make antigens accessible to antibodies [7]. Of note, fixation of organoids with methanol does not require antigen retrieval, yet our prior staining showed that this method is inappropriate for detecting some epitopes and the image quality is lower when compared to fixation performed with formaldehyde. Second, due to their compact shape and thickness, several antibodies fail to penetrate intact organoids. This can be worked around using detergents in the staining protocol. Third, light scattering and/or absorption by biological tissues is an obstacle to in depth imaging. However, optical clearing compounds are now available to render tissues transparent and straighten the light path, improving the thickness of samples that can be imaged in their entirety. Finally, when dispersed in a Matrigel dome, organoids are sometimes out of reach when using lenses with a magnification higher than x10. On the opposite, mounting organoids on a slide with a coverslip at the end of the staining leads to the flattening of the organoids and prevents from imaging them in their native shape.

Here we propose an easy workflow allowing (1) immunostaining with a HIER step, (2) preservation of the intact shape of the enteroids, (3) sample immobilization in a focal plane reachable for high resolution/short working distance lenses, and (4) minimizing the risk of loss of precious material (Fig 1).

## Methods

The protocol described in this peer-reviewed article is published on protocols.io (https://dx.doi.org/10.17504/protocols.io.kqdg323m7v25/v1) and is included for printing purposes as S1 File.

### Ethics and study population

The study protocol conforms to the ethical guidelines of the 1975 Declaration of Helsinki and was approved by the institution's human research and ethical committee (Comité Ethique Hospitalo-facultaire, Cliniques Universitaires Saint Luc, N° B403201422657). Written informed consent was obtained from all patients and controls. Patients hospitalized for selective alcohol withdrawal in a dedicated alcohol withdrawal unit and controls undergoing out-

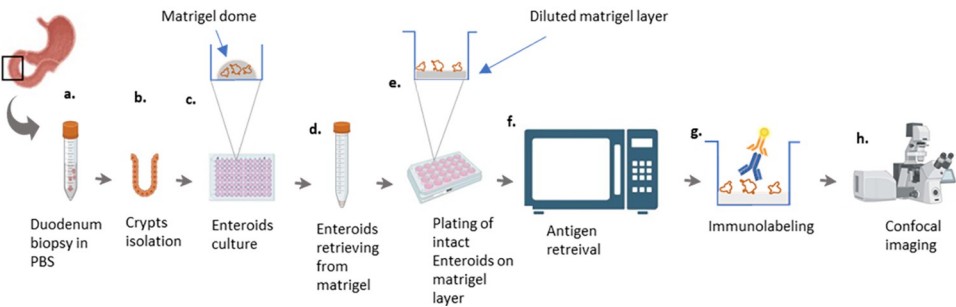

**Fig 1. Workflow of enteroid formation, staining and imaging.** Crypts are isolated (b) from duodenum-derived biopsies (a) and cultured in a Matrigel dome (c). Whole enteroids are retrieved from Matrigel after growth (d) and transferred to Matrigel-coated 24 well plates (e). After attachment, enteroids are fixed and submitted to microwave antigen retrieval (f) followed by immunolabeling (g). After clearing, images are acquired by a confocal microscope (g).

patient upper gastro-intestinal endoscopy for dyspepsia or reflux symptoms were recruited between 1st November 2023 and 30th July 2024. Control patients were only retained for the study if they had a normal gastro-intestinal endoscopy and no histological changes on the biopsy samples.

## Crypt isolation ○ timing 3h

All the following steps are performed at 4°C on ice

- Collect duodenum biopsies in sterile PBS (without Ca2+ or Mg2+)

- Wash 3x2 min under mild rotation in PBS + antibiotics and antimycotic 100X (Invitrogen, 15240062)

- Incubate for 1h in 2mM EDTA in a 15 ml conic tube under slow rotation

- This step aims to loosen the crypts from the intestinal mucosa.

- Allow biopsies to settle down (10 sec) and discard the supernatant

- Add 1 ml dissociation buffer (43.4 mM sucrose + 54.9 mM D-sorbitol in DPBS) and transfer into 2 ml cryotube

- This step allows trapping of ions involved in molecular enzymatic degradation of the crypts

- Shake vigorously by hand. Allow biopsies to settle down (30 seconds) and collect the supernatant containing crypts in a 15 ml conic tube. The remaining biopsies will sediment in the cryotube.

- Repeat the previous step twice or three times to ensure total recovery of crypts

- Add 5 ml of DMEM medium supplemented with 1% Penicillin/Streptomycin and 20% FBS

- Centrifuge at 400 g for 8 min at 4°C

- Discard the supernatant

- Wash once in 2 ml DMEM medium

- Filter through 70 μm cell strainer

- Centrifuge at 400 g for 7 min at 4°C

- Discard the supernatant

- Resuspend the pellet in 20–30 μl cold Corning® Matrigel® Growth Factor Reduced (GFR) Basement Membrane Matrix, Phenol Red-free (VWR, 734–1101) and plate 25–30 μl per well in flat bottom 96 well plate (96 well plates, Greiner [(G)] Bio-one, CELLSTAR ® cat N˚ 655180)

- Incubate at 37˚C, 5% CO2 for 10 min to allow the Matrigel to polymerize

- Add 37˚C heated IntestiCult OGM Human (STEMCELL technologies, #06010) containing 1% Penicillin/Streptomycin

## Enteroid passaging and culture ○ timing 7-14d

- Remove medium from the Matrigel dome

- Add 200 μl/well of Corning® Cell Recovery Solution (Corning, #734–0107)

- Incubate on ice for 40 min

- Pull the enteroids into a 15 ml conic tube

- Centrifuge at 400 g for 7 min

- Discard the supernatant

- Wash once with 2 ml cold DMEM medium

- Centrifuge 400 g for 7 min at 4˚C

- Discard the supernatant

- Add 2 ml Gibco™ TrypLE™ Express Enzyme (1X), phenol red (Thermofisher, #12605010) to the pellet

- Incubate for 25 min at 37˚C

- Neutralize with 3 ml DMEM medium containing FBS (1/20)

- Pipette up and down 10 times to dissociate remaining enteroids clusters

- Centrifuge as previously and discard the supernatant

- Wash once with DMEM

- Resuspend the pellet in cold Matrigel and plate approximately 15000 single cells in 20–30 μl per well in a flat bottom 96 well plate (Greiner [(G)] Bio-one, CELLSTAR ® cat N˚ 655180).

  **#TIPS** All enteroids are not dissociated into single cells. Some cluster or 3–5 cells could remain

- Add heated IntestiCult OGM Human (STEMCELL technologies, #06010) containing 1% Penicillin/Streptomycin

- Change medium every 2 days until day 6

- Change medium every day from day 7

## Enteroid preparation for imaging

**Plate coating ○ timing 30 min.** ■ Dilute Matrigel 50x with cold PBS

- Plate 200 μl/well of diluted Matrigel in a 24 well plate (Greiner [(G)] Bio-one, CELLSTAR ⓡ cat N˚ 62210)

    ▲ **CRITICAL STEP** A thin homogeneous coating of Matrigel is critical to allow enteroids attachment in a similar plane.

    ▲ **CRITICAL STEP** If high resolution is needed, select plates with coverslip bottom

- Incubate at least 20–30 min at 37˚C to allow the Matrigel to polymerize

## Enteroids retrieving ○ timing 3h

- Remove IntestiCult OGM Human culture medium from the enteroids

- Add 200 μl/well of Corningⓡ Cell Recovery Solution (CRC, 354253)

- Place the plate on ice

- Incubate for 40 min on ice

- Transfer the detached enteroids in a 15 ml conic tube

- Wash each well 2X with Cell Recovery Solution and add this to the 15 ml tube

- Centrifuge at 400 g for 7 min at 4˚C

- Discard the supernatant

- Wash 1x with DMEM medium

- Centrifuge as previously and discard the medium

- Add 200 μl/well culture medium on each pellet (containing treatment if applicable)

\# **TIPS** Calculate the adequate volume depending on the total number of wells

- Remove the 24 well coated plate from the incubator and remove the excess PBS with 200 μl tips

- Plate 200 μl/well of cell suspension

- Incubate 1.5-2h at 37˚C to allow the enteroids to attach to the plate

## Fixation ○ timing 1h

- Remove the excess of medium

- Wash 1x with heated PBS (37˚C)

- Fix with cold 4% formaldehyde for 45 min at 4˚C

- Wash 1x with cold PBS (4˚C)
    \# **TIPS** Samples can be stored in PBS at 4˚C, before being processed
    ! **CAUTION** Manipulate the formaldehyde under the hood.

- Proceed directly to permeabilization if the antibody does not require an antigen retrieval step

## Antigen retrieval, permeabilization, blocking ○ timing 1 day

- Remove PBS

- Prepare 250 ml of citrate buffer:

  add 25mL of citrate buffer stock solution 10x concentrated (3.78g citric acid and 24.12g sodium citrate dihydrate in 100ml distilled water, pH adjusted at 5.6, volume adjusted at 1l, storage at 4˚C) and 675µL of 20% Triton-X100 solution (diluted in distilled water) in 225mL distilled water

- Fill each well with citrate buffer (1x) at RT

- Boil the remaining buffer in microwave at 900 Watt and keep it aside

- Put the plate in the microwave

- Heat at 900W until boiling (approximately 30 sec)

- Replace the citrate buffer by hot citrate buffer

- Repeat three times by filling the well before each boiling

- Heat at 90 Watt for 15 min

- Boil 3 times at 900 Watt

- Let the plate cool down for 10-15min

- wash 1X with distilled water

- Permeabilize and Block with Organoid Washing Buffer (OWB) overnight (Dekkers et al., 2019)

  - OWB : 1l of PBS containing 1 ml of Triton X-100 and 2g of BSA

## Antibody staining ○ timing 2 days

- Dilute primary antibody (Table 1) in OWB solution

- Incubate enteroids with 300 µl of primary antibody solution overnight at 4˚C

- Wash 2x 5 min with 500 µl OWB

- Incubate with 300 µl of secondary antibody (Table 1) in OWB solution overnight at 4˚C

- Wash twice with OWB as described previously

- Incubate with 300 µl of DAPI (SIGMA D9542, stock 1µg/ml diluted 1:1000) for at least 1h

**Table 1. Antibodies.**

| Antibodies | Abbreviation | Dilution | Concentration | Species | Supplier | Catalogue n˚ | RRID |
|---|---|---|---|---|---|---|---|
| Mucin2 | MUC2 | 1:200 | 1 µg/ml | Mouse | Santa Cruz | Sc-515032 | AB_2815005 |
| Lyzozyme | LYZ | 1:400 | 0.625 µg/ml | Rabbit | Abcam | Ab223503 | |
| Olfactomedin 4 | OLFM4 | 1:200 | 0.035 µg/ml | Rabbit | Cell Signaling | 14369S | AB_2798465 |
| E-cadherin | E-cad | 1:100 | 665 µg/ml | Mouse | Dako | M3612 | |
| Zonula Occludens-1 | ZO-1 | 1:100 | 2.5 µg/ml | Rabbit | Invitrogen | 617300 | |
| Alexa FluorTM 488 donkey anti-mouse | Anti-Ms | 1:1500 | 1.33 µg/ml | Donkey | Invitrogen | A21202 | |
| Alexa FluorTM 594 donkey anti-rabbit | Anti-Rb | 1:1500 | 1.33 µg/ml | Donkey | Invitrogen | A21207 | |

List of antibodies with their corresponding concentration and dilution.

- Wash twice with OWB

- Wash with distilled water

- Remove completely the water

## Clearing and imaging

- Add 3 drops/well of RapiClear 1.49 (Sunjin Lab #RC147001)

  ▲ **CRITICAL STEP** Without clearing, only the external cell layers are detectable.

- Image after 2h or the next day (Figs 2–4)

## Results and discussion

### Results

See Figs 2–4.

### Discussion

In order to assess the cell components of intestinal enteroids, we performed immunostaining to detect MUC2, LYZ, OLFM4, and ZO-1 proteins, respectively expressed by goblet, paneth, stem and stem-like cells and tight junctions. Entire enteroids were removed from Matrigel domes and were plated on a diluted Matrigel layer to immobilize them on a single plate for further imaging. In this protocol, the pre-formed enteroids are not embedded in the Matrigel and hence are directly accessible to antibodies with an accessible working distance from the objectives. Then, they were submitted to fixation and immunostaining without any prior antigen retrieval. They were finally imaged in a clearing solution with a confocal microscope.

Using this protocol, enteroids were attached to the plate without forcing or destroying their morphology. Furthermore, after their attachment and fixation, they were tightly stuck to the plate and did not detach during the HIER nor during the washing steps. Consequently, the

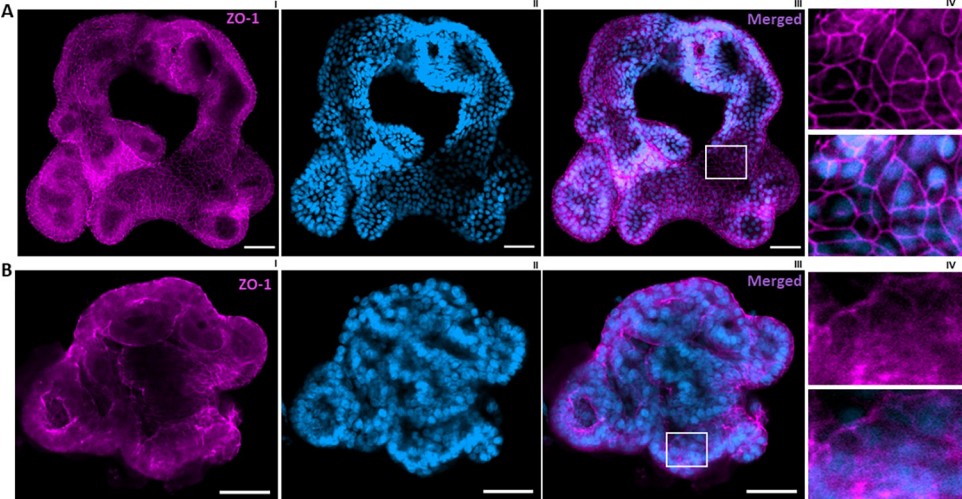

**Fig 2. Confocal images of enteroids on day 18.** Staining of tight junctions using Zona occludens 1 (ZO-1) in purple shows specific binding of the antibody to its epitope without antigen retrieval (A) but not with antigen retrieval (B). DAPI in blue stains the nucleus. AIV & BIV, optical section of indicated volume in AIII and BIII. Scale bar 50 μm.

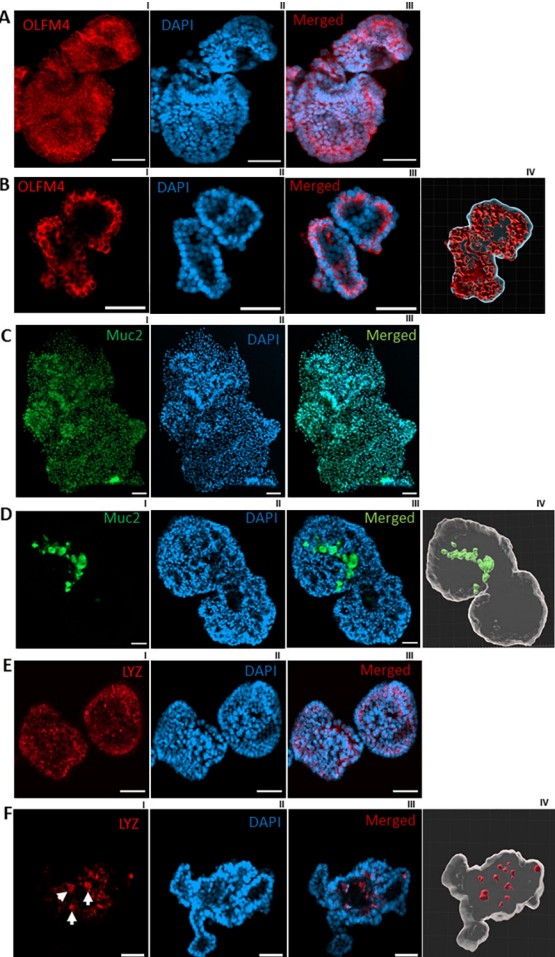

**Fig 3. Confocal imaging of enteroids on day 14.** A. OLFM4 (Olfactomedin 4) in red (A, B), Muc2 (Mucin 2) in green (C, D), Lyz (Lyzozyme) in red (E, F). Unspecific binding of OLFM4, MUC2, and LYZ respectively in (A), (C) and (E) without any prior antigen unmasking. B, D and F, specific binding of OLFM4, Muc2 and Lyz to stem cell and undifferentiated cells, Goblet cells and Paneth cells respectively after antigen retrieval. BIV., DIV and FIV, 3D representations of figures BIII, DIII and FIII respectively. Scale bar 50 μm.

number of enteroids before and after the staining was approximately the same, thus avoiding excessive loss of material.

The cellular localization of a protein of interest is an important element to assess the specificity of an immunostaining. The results represented in Fig 2 showed satisfactory tight junction protein ZO-1 detection. However, OLFM4, MUC2 and LYZ staining showed unspecific binding (mainly in the nucleus) of their respective antibodies.

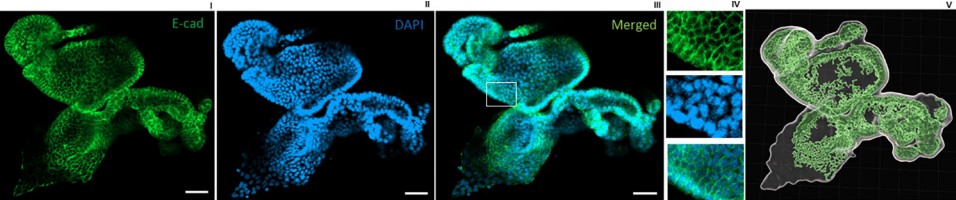

**Fig 4. Confocal imaging of enteroids submitted to antigen retrieval on day 14.** E-cad (membrane protein) in green, marker of epithelial cells. IV., optical sections of indicated volume in III. 3D representation IV. Scale bar 50 μm.

We hypothesized that the lack of specific staining was due to inaccessibility of the antigens. As an antigen retrieval process, we first tried the FLASH technique proposed in a recent publication [8]. Although this approach might be successful for labeling of some epitopes, it did not allow to detect MUC2, LYZ and OLFM4 proteins in intestinal enteroids. However, as illustrated in Fig 3, we obtained specific staining of Muc2, OLFM4, LYZ protein in the enteroids using our protocol of convenient sample preparation while avoiding sample loss and imaging at the same focal length range. Compared to a traditional Immunostaining workflow on 2D sections of spheroids, we could avoid some challenging steps including: 1) the difficulty of finding organoids when sectioning, 2) time consuming 3) the lack of determining the level of the organoid/spheroid sectioned, 4) the limitation to acquiring only one layer [9].

These results support the added value of our protocol and indicate the need of appropriate antigen retrieval for detecting some epitopes by whole mount immunostaining [10–12].

## Conclusion

The technique suggested in this protocol allows enteroids to attach to the plate while keeping their shape and structure. After fixation, the attached enteroids are not removed throughout the whole staining process including Antigen retrieval and washing steps. Moreover, imaging entire enteroids by acquiring the different Z-stack positions prevents losing information about each single enteroid.

The advantage/ease of this technique is the use of a single plate with different experimental conditions, and we could use as low as 10 enteroids for the staining process. Moreover, no centrifugation process is required, and the washing steps are straight forward.

## Limitations

Fixation of the enteroids on a plate does not allow Light-sheet microscopy since the enteroids are strongly attached to the plate and cannot be removed after the staining. Moreover, Imaging resolution is limited to 10X microscope lenses, and 96 well-plates are not suitable for this technique. Additionally, this protocol has been used only for enteroids even though it could be applied to other types of organoids.

## Reagents

### Dissociation buffer

Prepare the dissociation buffer by mixing 43.4 mM Sucrose solution (7.43 g in 500 ml distilled water, MW 342.3) and 54.9 mM sorbitol solution (5 g in 500 ml milli Q).

### EDTA 2mM pH 8

Prepare 0.5 M stock solution by adding 93.05 g of ETDA (MW 372.2) to 500 ml of distilled water. Adjust pH to 8. Dilute 250X the stock solution for ready-to-use solution.

### Citrate buffer

For Antigen retrieval, prepare a 60x concentrated stock solution of citrate buffer with 9.15 g of citric acid (MW: 210.1), 70.35 g tri-sodium dihydrate (MW: 294.1) and 2.4 g Trisma Base (MW: 121.14) in 1L. Adjust pH to 5.7. storage at 4˚C.

For ready-to-use solution, add 25ml of citrate buffer stock solution in 1475 ml milli Q water. Add 675 μl of 20X triton X-100 (4 ml of Triton 100 in 16 ml Milli Q water) for each 250 ml 1X citrate buffer. Storage at

### PBS-Tween20

Prepare 0.1% (vol/vol) Tween 20 in PBS by adding 50 µl of Tween-20 in 50 ml PBS.

### PBS-BSA 1%

Add 0.5g of BSA in 50 ml PBS.

### Blocking and washing buffer

Prepare Organoid Washing Buffer with PBS containing 0.2% BSA and 0.1% Triton X-100 by adding 100 mg BSA and 50 µl Triton X-100 in 50 ml PBS.

## Equipments

### Consumables

- 96 well plates, Greiner [G] Bio-one, CELLSTAR Ⓡ cat N˚ 655180

- 24 well plates, Greiner [G] Bio-one, CELLSTAR Ⓡ cat N˚ 62210

- 15 ml conic tubes, Greiner [G] Bio-one, CELLSTAR Ⓡ 88271N

### Bench apparatus

- Microwave oven

- Incubator

- Centrifuge

### Microscopes

The images were acquired by a Zeiss LSM800 inverted confocal microscope equipped with 4 lasers (405, 488, 561 and 640 nm), Variable Secondary Dichroics, GaAsp detectors and x10/ NA 0.3 Plan-Apochromat lens.

### Software

Zen 3.5 (blue edition) and Imaris (Bitplane) were used for 3D reconstitution of the images acquired by a confocal microscope.

## Supporting information

**S1 Fig. Confocal images of Muc2 staining showing the different acquisition layers (z-stacks).**
(TIF)

**S2 Fig. Confocal images of OLFM4 staining showing the different acquisition layers (z-stacks).**
(TIF)

**S3 Fig. Confocal images of Muc2 staining.**
(TIF)

**S1 File. Step-by-step protocol, also available on protocols.io.**
(PDF)

## Author Contributions

**Conceptualization:** Ami Gloria Toulehohoun.

**Data curation:** Ami Gloria Toulehohoun.

**Formal analysis:** Ami Gloria Toulehohoun, Caroline Bouzin.

**Methodology:** Ami Gloria Toulehohoun, Aurélie Daumerie, Luca Maccioni.

**Software:** Ami Gloria Toulehohoun, Caroline Bouzin.

**Supervision:** Peter Stärkel.

**Validation:** Caroline Bouzin, Peter Stärkel.

**Visualization:** Caroline Bouzin, Peter Stärkel.

**Writing – original draft:** Ami Gloria Toulehohoun.

**Writing – review & editing:** Caroline Bouzin, Peter Stärkel.

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
