## [Decision Letter · Decision Letter 0]

17 Oct 2024

PONE-D-24-359333D fluorescence staining and imaging of low amount of organoids: protocol accessible to allPLOS ONE

Dear Dr. TOULEHOHOUN,

Thank you for submitting your manuscript to PLOS ONE. After careful consideration, we feel that it has merit but does not fully meet PLOS ONE’s publication criteria as it currently stands. Therefore, we invite you to submit a revised version of the manuscript that addresses the points raised during the review process.

**ACADEMIC EDITOR: **

**Please follow all reviewers comments and reply for their questions. **

We look forward to receiving your revised manuscript.

Kind regards,

Ayman A. Swelum

Academic Editor

PLOS ONE

4. We note you have not yet provided a protocols.io PDF version of your protocol and/or a protocols.io DOI. When you submit your revision, please provide a PDF version of your protocol as generated by protocols.io (the file will have the protocols.io logo in the upper right corner of the first page) as a Supporting Information file. The filename should be S1_file.pdf, and you should enter “S1 File” into the Description field. Any additional protocols should be numbered S2, S3, and so on. Please also follow the instructions for Supporting Information captions [https://journals.plos.org/plosone/s/supporting-information#loc-captions]. The title in the caption should read: “Step-by-step protocol, also available on protocols.io.”

Please assign your protocol a protocols.io DOI, if you have not already done so, and include the following line in the Materials and Methods section of your manuscript: “The protocol described in this peer-reviewed article is published on protocols.io (https://dx.doi.org/10.17504/protocols.io.[...]) and is included for printing purposes as S1 File.” You should also supply the DOI in the Protocols.io DOI field of the submission form when you submit your revision.

If you have not yet uploaded your protocol to protocols.io, you are invited to use the platform’s protocol entry service [https://www.protocols.io/we-enter-protocols] for doing so, at no charge. Through this service, the team at protocols.io will enter your protocol for you and format it in a way that takes advantage of the platform’s features. When submitting your protocol to the protocol entry service please include the customer code PLOS2022 in the Note field and indicate that your protocol is associated with a PLOS ONE Lab Protocol Submission. You should also include the title and manuscript number of your PLOS ONE submission.

Reviewers' comments:

Reviewer's Responses to Questions

**Comments to the Author**

1. Does the manuscript report a protocol which is of utility to the research community and adds value to the published literature?

Reviewer #1: No

Reviewer #2: Yes

Reviewer #3: Yes

Reviewer #4: Yes

Reviewer #5: No

Reviewer #6: No

Reviewer #7: Yes

2. Has the protocol been described in sufficient detail?

To answer this question, please click the link to protocols.io in the Materials and Methods section of the manuscript (if a link has been provided) or consult the step-by-step protocol in the Supporting Information files.

The step-by-step protocol should contain sufficient detail for another researcher to be able to reproduce all experiments and analyses.

Reviewer #1: No

Reviewer #2: Yes

Reviewer #3: Yes

Reviewer #4: Yes

Reviewer #5: Yes

Reviewer #6: Yes

Reviewer #7: Yes

3. Does the protocol describe a validated method?

Reviewer #1: No

Reviewer #2: Yes

Reviewer #3: Yes

Reviewer #4: Yes

Reviewer #5: Yes

Reviewer #6: Yes

Reviewer #7: No

4. If the manuscript contains new data, have the authors made this data fully available?

Reviewer #1: No

Reviewer #2: Yes

Reviewer #3: No

Reviewer #4: Yes

Reviewer #5: No

Reviewer #6: N/A

Reviewer #7: N/A

**5. Is the article presented in an intelligible fashion and written in standard English?**

Reviewer #1: **No: **must need proof reading services

Reviewer #2: Yes

Reviewer #3: Yes

Reviewer #4: Yes

Reviewer #5: Yes

Reviewer #6: **No: **There are spelling mistakes and English grammar errors. Editing is recommended.

Reviewer #7: Yes

6. Review Comments to the Author

Reviewer #1: The manuscript was poorly written, and the authors failed to convince that the method they developed was reliable and repeatable based on what was presented here. The heat-induced method of epitope retrieval was a well-established technique routinely used for histological staining. The sandwich method of placing organoids and paraformaldehyde fixation is recommended by Corning and routinely used in organoid labs. Hence, I do not see novelty in their study. In the abstract, the authors mentioned that 3D imaging is necessary but failed to show 3D imaging techniques such as Light-sheet microscopy. I think the authors should have at least shown a confocal-built 3D image. It is well known, and even described in the Matrigel manual, that the proteins in Matrigel can autofluoresce. Lastly, when the authors claim that they developed an easy workflow for staining, they should have compared their method with a more difficult workflow and summarized their results in a table/figures.

Comments:

1. Line 28: “mimicking patient…”, why not animals? Organoids were generated in animals; re-phrase this statement.

2. Figure 3: I can't find all the images described in the figure legend

3. Table 1 should include conc. , not dilutions of antibodies.

4. Must add more references for some of the statements quoted, e.g. lines 54-55

5. Line 139: what is 1/200?

6. Line 143: what is a chelation buffer? I think there may be additional components in this buffer.

7. Line 146: “…min)?

8. Line 200: Which culture medium?

9. Line 214: Once Matrigel is polymerized, how can the organoids attach?

10. Line 270: Compared to which techniques??

11. It looks like a copy-paste lab protocol, not proofread

12. Inconsistency in describing the vendors or manufacturer throughout the manuscript. Corning is the manufacturer, not VWR

13. It is interesting to see only one manuscript referenced from Dr. Clever's lab

14. This manuscript requires thorough proofreading for grammar, spell-checks, font sizes, and inconsistency.

Reviewer #2: This protocol described how to stain and image 3D organoids. Some questions are:

1. Please state the confocal microscope in the title, since 3D imaging could be archieved by other ways, for example fluorescent MOST.

2. In looking at the presented images (2-4), it seems that they are not like 3D imaging, because they do not show the whole organoids. Did the authors only image a few layers, not the whole organoids?

3. How low is low amount?

4. Please state the types of organoids. Since they are from crypts, they should be intestine organoids. Then could this protocol be used to other organoids, such as brain organoids, since brain organoids contain more contact core.

Reviewer #3: The protocol is easy to comprehend and follow, but requires some fine tuning to be more applicable for efficient visualization at a higher magnification.

These are some minor comments:

There are some incomplete sentences in the methods (L141 and L146). Please revise and double-check.

For each image and in the protocol, please specify on which day of culture you sampled the organoids (day 7 till day 14).

Throughout the figures, there are inappropriate staining for DAPI. The nuclear boundaries are not clear. Do you have any explanation for this?

Also, for MUC2, it is restricted to particular pole of the organoids, why it is not homogenous throughout the lumen?

The same for LYZ, it appeared as you reduced the brightness and the images will be different when increasing the brightness.

Reviewer #4: The protocol presents an interesting and accessible option for researchers. While the manuscript offers valuable insights, including comparisons with the FLASH method, a more comprehensive evaluation is needed for a deeper understanding of its effectiveness. Comparing it with traditional techniques will further clarify its practical applications.

To strengthen the manuscript, I recommend adding comparisons between this protocol and results from traditional, slower methods, alongside FLASH. This will help: provide essential context on the efficacy and reliability of your approach; highlight potential advantages and limitations, particularly regarding resolution and structural preservation. Visual comparisons across different methods will enhance understanding, and incorporating established protocols will better situate your findings within existing literature, bolstering the study’s credibility. Furthermore, it would be beneficial to explore whether this method is applicable to organoids other than those derived from the duodenal mucosa.

Reviewer #5: Dear author the protocol presented add the microwave to demask antigens that could be a TIP to improve the antigen-antibody recognizon for this reason the protocol is not particular original

could the authors add additional use of this protocol not only for imaging

however some steps need to be clarify such as The step with citrate buffer at RT . Is the working solution at this step 10X?

Does Organoid Washing Buffer contain 1ml of Triton 100% with 2g BSA? please indicate in the step

Lysozymes and MUC2 are not present images these markers are important because are intracellular to show if the antibody is able to go inside the cell

Reviewer #6: Re: 3D Fluorescence Staining and Imaging of Low Amount of Organoids: Protocol Accessible to All

Organoids are miniature tissues that exhibit the structure and specific functions of various organs. In recent years, with the advancement of organoid production techniques, these models have significantly contributed to the study and understanding of the mechanisms involved in the pathogenesis of various diseases. The ability to visualize organoids using three-dimensional fluorescence techniques in many cases, and the preparation of these organoids for imaging, are of crucial importance. The researchers who developed this protocol noted that in a microwave-based antigen retrieval method, which they term HIER, the target antigenic epitopes are unmasked, facilitating easier and more efficient labeling of the cells that compose the organoids. However, it is important to note that the microwave antigen retrieval method is not a novel technique, and it has not yielded consistent results across all antibodies they aimed to demonstrate. Additionally, I have further critiques and suggestions as outlined below.

Main Comments:

The background/introduction section of the protocol paper lacks sufficient citations. Certain statements sharing information and research results are not attributed to any specific study, despite not being original findings of this work.

In your results, you stated that no data were shown for MUC2, LYZ, and OLFM4 due to non-specific staining in intestinal organoids. However, OLFM4 staining is visible in Fig. 3. Furthermore, the figure legend of Fig. 3 indicates that OLFM4 is labeled in red in panels A and B, MUC2 is labeled in green in panels C and D, and LYZ is labeled in red in panels E and F. However, the figure file only contains panels A and B, where OLFM4 staining is visible. This discrepancy needs to be addressed and clarified.

Why did you perform imaging on a confocal microscope after tissue clearing? The resolution still appears to be low. Using light-sheet microscopy instead could improve image quality and better highlight the key steps in your protocol.

It is recommended to share the key parameters (e.g., exposure time, gain, etc.) used in the confocal imaging process.

Additionally, when referring to intestinal organoids, it is suggested to indicate "enteroids" in parentheses upon the first mention and to consistently use "enteroids" in subsequent sections.

Reviewer #7: The protocol is generally clear and well-written. The only major request I have is to indicate how many organoids have been successfully stained with this protocol.

Has this protocol been used in a published work?

Plaese note that on line 146 timing is missing.

Please carefully check English spelling, e.g. line 17: “lost” should be replaced with “loss”, line 57:”brake” should be replaced with “break”

7. PLOS authors have the option to publish the peer review history of their article (what does this mean?). If published, this will include your full peer review and any attached files.

Reviewer #1: No

Reviewer #2: No

Reviewer #3: **Yes: **Islam M. Saadeldin

Reviewer #4: No

Reviewer #5: No

Reviewer #6: **Yes: **Dr. Özgecan Kayalar (MSc., PhD.)

Reviewer #7: No

---

## [Author Response · Author response to Decision Letter 0]

30 Nov 2024

Dear reviewers,

We thank you for your insightful comments on the protocol. This has added a great value to the content of the manuscript. The revisions were made according to your remarks.

Best regards,

Ami Gloria Toulehohoun

---

## [Editor Report · Decision Letter 1]

4 Dec 2024

3D fluorescence staining and confocal imaging of low amount of intestinal organoids (enteroids): protocol accessible to all

PONE-D-24-35933R1

Dear Dr. Ami Gloria TOULEHOHOUN,

We’re pleased to inform you that your manuscript has been judged scientifically suitable for publication and will be formally accepted for publication once it meets all outstanding technical requirements.

Kind regards,

Ayman A. Swelum

Academic Editor

PLOS ONE

---

## [Editor Report · Acceptance letter]

23 Dec 2024

PONE-D-24-35933R1 

PLOS ONE

Dear Dr. TOULEHOHOUN, 

I'm pleased to inform you that your manuscript has been deemed suitable for publication in PLOS ONE. Congratulations! Your manuscript is now being handed over to our production team.

Kind regards, 

on behalf of

Professor Ayman A. Swelum 

Academic Editor

PLOS ONE